# Mitochondrial Aldehyde Dehydrogenase 2 Represents a Potential Biomarker of Biochemical Recurrence in Prostate Cancer Patients

**DOI:** 10.3390/molecules27186000

**Published:** 2022-09-15

**Authors:** Dechao Feng, Weizhen Zhu, Jia You, Xu Shi, Ping Han, Wuran Wei, Qiang Wei, Lu Yang

**Affiliations:** Department of Urology, Institute of Urology, West China Hospital, Sichuan University, Chengdu 610041, China

**Keywords:** mitochondrial aldehyde dehydrogenase 2, prostate cancer, biochemical recurrence, tumor immune microenvironment, competing endogenous RNA network

## Abstract

Background: We aimed to explore the role of mitochondrial aldehyde dehydrogenase 2 (ALDH2) in prostate cancer (PCa) patients and provide insights into the tumor immune microenvironment (TME) for those patients undergoing radical radiotherapy. Methods: We performed all analyses using R version 3.6.3 and its suitable packages. Cytoscape 3.8.2 was used to establish network of competing endogenous RNAs (ceRNAs). Results: Downregulation of ADLH2 was significantly associated with higher risk of BCR-free survival (HR: 0.40, 95%CI: 0.24–0.68, *p* = 0.001) and metastasis-free survival (HR: 0.21, 95%CI: 0.09–0.49, *p* = 0.002). Additionally, ALDH2 repression contributed to significantly shorter BCR-free survival in the TCGA database (HR: 0.55, 95%CI: 0.33–0.93, *p* = 0.027). For immune checkpoints, patients that expressed a higher level of CD96 had a higher risk of BCR than their counterparts (HR: 1.79, 95%CI: 1.06–3.03, *p* = 0.032), as well as NRP1 (HR: 2.18, 95%CI: 1.29–3.69, *p* = 0.005). In terms of the TME parameters, the spearman analysis showed that ALDH was positively associated with B cells (r: 0.13), CD8+ T cells (r: 0.19), neutrophils (r: 0.13), and macrophages (r: 0.17). Patients with higher score of neutrophils (HR: 1.75, 95%CI: 1.03–2.95, *p* = 0.038), immune score (HR: 1.92, 95%CI: 1.14–3.25, *p* = 0.017), stromal score (HR: 2.52, 95%CI: 1.49–4.26, *p* = 0.001), and estimate score (HR: 1.81, 95%CI: 1.07–3.06, *p* = 0.028) had higher risk of BCR than their counterparts. Our ceRNA network found that PART1 might regulate the expression of ALDH via has-miR-578 and has-miR-6833-3p. Besides, PHA-793887, PI-103, and piperlongumine had better correlations with ALDH2. Conclusions: We found that ALDH2 might serve as a potential biomarker predicting biochemical recurrence for PCa patients.

## 1. Background

The global population of people over the age of 65 is growing at an unprecedented rate, with 1.6 billion expected by 2050 [1]. Prostate cancer (PCa) is the most common urological malignancy in men with high incidence over the age of 65 years, and belongs to the fifth leading cause of cancer death among men globally [2]. This disease affects millions of men worldwide, and the problem will only worsen as the world’s population ages.

Increasing evidence suggests that there are close correlations between health dimensions and aging phenotypes, particularly autophagy, mitochondrial dysfunction, cellular senescence, and DNA methylation [1,3]. Warburg discovered aerobic glycolysis in tumor tissue in 1924, and de novo lipid synthesis in neoplastic tissue was discovered in 1953 [4,5]. A large number of studies on fatty acid metabolism contributed to one of the most common cancer hallmarks in 2011, namely “reprogramming of energy metabolism” [4,6]. Changes in intracellular and extracellular metabolites that can occur as a result of cancer-related metabolic reprogramming have a significant impact on gene expression, cellular differentiation, and the tumor microenvironment [7]. Through the generation of building blocks for membrane synthesis, the provision of substrates for ATP synthesis, and the regulation of signaling pathways involved in cell proliferation and survival, fatty acids play an important role in tumor initiation, development, and disease progression in caners [4].

The study of metabolic abnormalities in tumors has increased dramatically over the last decade. The complex interactions within the tumor microenvironment and adjacent stroma frequently influence how cancer cells use lipids [8]. Mitochondrial aldehyde dehydrogenase 2 (ALDH2), a gene related to fatty acid metabolism, has been demonstrated to be highly associated with prognosis and chemoradiotherapy sensitivity of many cancers, including leukemia, renal cell carcinoma, head and neck cancer, esophageal cancer, bladder cancer, hepatocellular cancer, pancreatic cancer, and ovarian cancer [9,10,11,12,13,14,15,16,17]. ALDH2 is made up of four identical subunits, each with three structural domains: the catalytic domain, the coenzyme structure nicotinamide adenine dinucleotide (NAD), and the oligomerization domain [18]. Fatty acid synthesis requires large amounts of nicotinamide adenine dinucleotide phosphate (NADP), an essential cofactor for biosynthetic reactions [4]. A growing body of evidence suggests that NAD (including NAD+ and NADH) and NADP (including NADP+ and NADPH) are fundamental common mediators of a wide range of biological processes, including energy metabolism, mitochondrial functions, calcium homeostasis, antioxidation/oxidative stress generation, gene expression, immunological functions, aging, and cell death [19]. Overproduction of reactive oxygen species (ROS) and lipid peroxidation promote the tumorigenesis, and ALDH2 is associated with the detoxification of reactive aldehydes generated from ROS-mediated lipid peroxidation, such as 4-hydroxy-2-nonenal (4-HNE), malondialdehyde, and acrolein [18]. Our previous study indicated ALDH2 might involve in the tumorigenesis and tumor progression of urological cancers [19]. However, less known empirical research to date has focused on exploring relationships between ALDH2 dysregulation and the PCa risk. From the perspective of fatty acid metabolism, we integrated multiple datasets to explore the role of ALDH2 in the prognosis of PCa and provide insights into the tumor immune microenvironment (TME) for those patients undergoing radical radiotherapy, which might enlighten the effect of ALDH2 on tumor stemness and multidrug resistance in the future.

## 2. Methods

### 2.1. Data Preparation

Our study has been registered in the ISRCTN registry (No. ISRCTN11560295). Four datasets (GSE46602 [20], GSE32571 [21], GSE62872 [22], and GSE116918 [23]) including 816 samples were combined and eliminated batch effects, which could be seen in our previous study [24]. The TCGA database PCa data included 550 samples, and 430 samples with complete prognostic data were used to validate the GEO prognostic value (Appendix A). From the GEO and TCGA, we extracted messenger RNA (mRNA) and long non-coding RNA (lncRNA) expression. The fatty acid metabolism gene sets were obtained from the gene set enrichment analyses (GSEA) website (https://www.gsea-msigdb.org/, accessed on 11 November 2021) and confirmed in a previous study [25]. The differentially expressed genes (DEGs) related to fatty acid metabolism were obtained through the intersection of gene sets of fatty acid metabolism, TCGA database and three GEO datasets (GSE46602 [20], GSE32571 [21], and GSE62872 [22]). The DEGs were regarded as llogFCl ≥ 0.4 and *p*. adj. < 0.05. The 248 tumor samples in GSE116918 [23] were used to identify the prognosis-related genes (Appendix A). The primary outcome was biochemical recurrence (BCR)-free survival, and the secondary outcome was metastasis-free survival.

### 2.2. Gene Interaction and Functional Enrichment Analysis

We analyzed the potential genes that might interact with ALDH2 using GeneMania [26]. We screened the long noncoding RNA (lncRNA) associated with BCR-free survival and differentially expressed between tumor and normal samples. Subsequently, we constructed a network of competing endogenous RNAs (ceRNAs) using lncBase [27], and miRDB [28,29]. We divided the 248 tumor patients undergoing radical radiotherapy in GSE116918 [23] into high- and low-expression groups according to the median of ALDH2. We further conducted the gene set variation analysis (GSVA) with “c2.cp.kegg.v7.4.symbols.gmt” from molecular signatures database [30]. The minimum and maximum gene set were 5 and 5000, respectively. Subsequently, “wilcox.test” function was used to evaluate the difference of each pathway between high- and low-expression of ALDH2. The fold change was 1.5, and we considered *p*. adj. < 0.01 and false discovery rate < 0.01 as statistical significance.

### 2.3. TME Analysis and Drug Analysis

We looked at the relationship between ALDH2 and 46 different immune checkpoints, as well as the difference in expression between BCR and no BCR groups. A prognosis analysis of differentially expressed checkpoints was also performed. Using the Timer and Estimate algorithms, a similar analysis was performed for immune parameters [31,32]. We investigated ALDH2 drug sensitivity using GSCALite, which incorporated data from the cancer therapeutics response portal (CTRP) and the genomics of drug sensitivity in cancer (GDSC) [33]. ALDH2’s diagnostic ability for drug (cyclophosphamide) and radiation resistance was evaluated using GSE42913 [34] and GSE53902 [35], respectively.

### 2.4. Immunohistochemistry Analysis

We confirmed the differential expression of ALDH2 at the protein level through the immunohistochemistry of a pair of carcinoma and adjacent tissue in our hospital.

### 2.5. Statistical Analysis

All analyses were carried out using R (version 3.6.3) and the appropriate packages. The ceRNA network was built using Cytoscape 3.8.2 [36]. If the data did not fit the normal distribution, we used the Wilcoxon test. ALDH2’s prognostic value was evaluated using Kaplan–Meier survival analysis and the log-rank test. The statistical significance level was set to two-sided *p* 0.05. The following significance levels were assigned: ns, 0.05; *, 0.05; **, 0.01, and ***, 0.001.

## 3. Results

### 3.1. Clinical Values of ALDH2

A total of 9 DEGs were identified through the intersection of DEGs of TCGA database and GEO datasets, and gene sets of fatty acid metabolism (Figure 1A). Downregulation of ALDH2 was significantly associated with a higher risk of BCR-free survival (HR: 0.40, 95%CI: 0.24–0.68, *p* = 0.001; Figure 1B) and metastasis-free survival (HR: 0.21, 95%CI: 0.09–0.49, *p* = 0.002; Figure 1C). Besides, ALDH2 repression contributed to significantly shorter BCR-free survival in the TCGA database (HR: 0.55, 95%CI: 0.33–0.93, *p* = 0.027; Figure 1D). The differential expression of ALDH2 in GEO datasets and TCGA database was presented (Figure 1E–F). At the protein level, ALDH2 expression was consistent with the mRNA expression, and the expression of ALDH2 was in plasma in carcinoma and adjacent tissue (Figure 1G). In addition, we observed that ALDH2 was differentially expressed between tumor and normal samples in most cancers in the TCGA database through the pan-cancer analysis (Figure 1H). In the GSE116918 [23], we observed a decreasing trend of ALDH2 expression with an upgrade of T stage and Gleason score (Figure 1I,J). BCR patients had a significant lower expression level of ALDH2 than no BCR patients (Figure 1K). However, the diagnostic ability of ALDH2 mRNA expression was low in distinguishing BCR from no BCR patients (AUC: 0.652; Figure 1L). Similar results were observed in terms of Gleason score and T stage in the TCGA database (Figure 1M–N), and patients with positive lymphnodes had a significant lower expression level of ALDH2 than patients with negative lymphnodes (Figure 1O). Moreover, ALDH2 presented a highly diagnostic ability discriminating drug (AUC: 0.906, 95%CI: 0.743–1.000; Figure 1P) and radiation resistance (AUC: 0.827, 95%CI: 0.612–1.000; Figure 1Q) from sensitivity.

### 3.2. TME Analysis and Possible Mechanisms of ALDH2

For immune checkpoints, CD96, LAIR1, NRP1, and PDCD1LG2 were significantly expressed higher in BCR group (Figure 2A), and ALDH2 was significantly related to BTNL2 (r: −0.15), CD200 (r: 0.20), CD200R1 (r: −0.20), CD226 (r: −0.12), CD276 (r: −0.27), CD40LG (r: −0.13), CD44 (r: 0.32), CD47 (r: 0.30), CD70 (r: −0.19), CD80 (r: −0.15), CTLA4 (r: −0.17), HHLA2 (r: −0.24), ICOSLG (r: −0.15), KIR3DL1 (r: −0.26), KLRD1 (r: −0.14), LAIR1 (r: −0.14), NRP1 (r: −0.17), TIGIT (r: −0.20), TNFRSF14 (r: −0.18), TNFRSF8 (r: −0.18), TNFRSF9 (r: −0.18), TNFSF15 (r: 0.22), and TNFSF9 (r: −0.28) (Figure 2B). Patients that expressed a higher level of CD96 had a higher risk of BCR than their counterparts (HR: 1.79, 95%CI: 1.06–3.03, *p* = 0.032; Figure 2C), as well as NRP1 (HR: 2.18, 95%CI: 1.29–3.69, *p* = 0.005; Figure 2D). In terms of the TME parameters, neutrophils, stromal score, immune score, and estimate score were scored higher, and tumor purity was scored lower in the BCR group than the no BCR group (Figure 2E). The spearman analysis showed that ALDH was positively associated with B cells (r: 0.13), CD8+ T cells (r: 0.19), neutrophils (r: 0.13), and macrophages (r: 0.17) (Figure 2F). Patients with a higher score of neutrophils (HR: 1.75, 95%CI: 1.03–2.95, *p* = 0.038), immune score (HR: 1.92, 95%CI: 1.14–3.25, *p* = 0.017), stromal score (HR: 2.52, 95%CI: 1.49–4.26, *p* = 0.001), and estimate score (HR: 1.81, 95%CI: 1.07–3.06, *p* = 0.028) had higher risk of BCR than their counterparts (Fi. 2G–J). In addition, patients with lower tumor purity had a higher risk of BCR than those with higher tumor purity (HR:0.55, 95%CI: 0.33–0.93, *p* = 0.028; Figure 2K). Moreover, we observed that several genes (SIRT3, ENTPD5, GCSH, SLC25A16, TSR1 and FBP1) were predicted to interact with ALDH2 (Figure 2L).

We found that beta alanine metabolism, limonene and pinene degradation, valine leucine and isoleucine degradation and propanoate metabolism were upregulated in the group highly expressing ALDH2 (Figure 3A). We observed that PART1 significantly expressed higher in tumor group compared to normal group through GSE46602 [20], GSE32571 [21], and GSE62872 [22] (Figure 3B). In GSE116918 [23], BCR patients had lower expression of PART1 than no BCR patients (Figure 3C). Patients with lower expression of PART1 had a higher risk of BCR than those with higher expression of PART1 (HR: 0.46, 95%CI: 0.27–1.77, *p* = 0.004; Figure 3D). Our ceRNA network found that PART1 might regulate the expression of ALDH via has-miR-578 and has-miR-6833-3p (Figure 3E). Six potentially sensitive drugs (AZD7762, CHIR-99021, PHA-793887, PI-103, SNX-2112 and piperlongumine) were found through the intersection of GDSC and CTRP (Figure 3F), among which PHA-793887, PI-103, and piperlongumine had better correlations defined as coefficients larger than 0.1.

## 4. Discussion

Because of increased life expectancy, the world’s elderly population is rapidly growing. Cancer and aging are biological processes that are inextricably linked, and the older population has a higher incidence of PCa than the younger population [2]. Endogenous metabolic insults and exogenous factors accumulate over time, causing oxidative stress and DNA damage, and senescent cells accumulate during the aging process and exhibit a senescence-associated secretory phenotype, facilitating a tumorigenic niche [37]. Mitochondria are the primary source of energy in eukaryotes, and significant amounts of ROS are produced during oxidative phosphorylation for the production of adenosine triphosphate at complexes I and III of the electron transport chain [38]. Superoxide anion, hydrogen peroxide, hydroxyl radicals, singlet oxygen, and lipid peroxyl radicals are examples of ROS [39]. The majority of superoxide is decomposed by superoxide dismutases, and several ROS can cause peroxidation of polyunsaturated fatty acids (PUFAs) in cellular membranes, forming lipid hydroperoxides as primary products, 4-HNE being one of the most bioactive and well-studied lipid alkenals [40,41]. Additionally, 4-HNE adducts have been implicated in the development and progression of cancer in studies [42,43]. Furthermore, constitutive 4-HNE levels may protect cancer cells from oxidative damage by activating mitochondrial uncoupling proteins, which help to reduce excessive ROS production and oxidative damage in cancer [44]. Furthermore, malondialdehyde has been linked to prostate cancer, breast cancer, hepatocellular carcinoma, oral cancer, and other cancers [45,46,47,48]. Cancer cells can activate adipocytes to lipolyze their triglyceride stores, delivering secreted fatty acids to cancer cells for uptake via a variety of fatty acid transporters and as the cancer–stromal interactions worsen, fatty acids secreted into the microenvironment can influence infiltrating immune cell function and phenotype. Lipid metabolic abnormalities, such as increased fatty acid oxidation and de novo lipid synthesis, can give the tumor an advantage in resisting chemotherapeutic and radiation treatments, as well as alleviating cellular stresses involved in the metastatic cascade [8]. Fatty acid metabolism-related gene, ALDH2 is critical in the removal of endogenous aldehydes formed during oxidative metabolism [18]. Thus, ALDH2 dysregulation may play a role in cancer carcinogenesis and progression [9,10,11,12,13,14,15,16,17]. We found that downregulation of ALDH2 contributed to shorter BCR-free survival in PCa patients, which was confirmed by the TCGA database, which was consistent with previous studies. Furthermore, we discovered that patients with lower levels of ALDH2 had a higher risk of metastasis than their counterparts. We discovered that ALDH2 was closely associated with many metabolism pathways, including fatty acid metabolism and PCa, in the GSEA analysis, confirming the importance of ALDH2 in PCa.

We discovered that PCa patients with higher neutrophil levels had shorter BCR-free survival than those with lower neutrophil levels in this study. We considered two possible explanations here: (1) neutrophils inhibited tumor growth by directly releasing cytotoxic mediators [49] and producing cytokines that promote the activation and proliferation of various immune cells [50], particularly T cells [51]. However, because immune repression was caused by metabolic competition of cancer cells, the killing effect of the recruited neutrophils on tumor cells was weakened, and BCR occurred again; (2) immune dysfunction may result in an imbalance in the type and number of neutrophils. Neutrophils of type N2 predominate in the number of neutrophils, which can promote cancer cell proliferation, migration, and invasion [52]. Furthermore, we discovered that the BCR group had higher stromal, immune, and estimate scores, while the no BCR group had lower tumor purity. Higher former scores were associated with a higher risk of BCR-free survival, whereas tumor purity had the opposite effect. In gastric cancer [53], lung adenocarcinoma [54], renal cell carcinoma [55], and colon cancer [56], stromal and immune scores were used to stratify prognosis. Estimate score was also linked to clinical endometrial cancer characteristics [57]. We hypothesized that these scores could be used as a prognostic indicator in PCa. Previous research has found that low tumor purity is associated with a poor prognosis and an immune-evasion phenotype in gastric cancer [58], and most recognized prognostic indicators are no longer significantly effective under different conditions of tumor purity [59], implying that tumor purity may play an important role in PCa treatment and prognosis assessment. Furthermore, we discovered that the expression of ALDH2 was positively correlated with CD8 + T cells, B cells, neutrophils, and macrophages. For CD8+ T cells, low ALDH2 expression resulted in acetaldehyde accumulation, which decreased the expression of PA28 and immune proteasome subunits and further inhibited CD8+ T cell activation by impairing the JAK/STAT1/IFNg signaling pathway [60]. Accumulation of acetaldehyde may also promote T cell inactivation by inhibiting AKT phosphorylation, glucose transporter 1 mRNA expression, and hypoxia-inducible factor-1a expression [61]. Intriguingly, metabolic competition between cancer cells and immune cells inhibited immune cell function, and metabolic reprogramming was also important in suppressing the immune attack on tumor cells and resistance to therapies [49]. In addition, we discovered that ALDH2 had a high diagnostic accuracy in distinguishing drug and radiation resistance from sensitivity in our study. In terms of immune checkpoints, CD96 and NRP1 were significantly higher expressed in BCR group and were related to worsen prognosis in PCa patients. Both CD96 and NRP1 prevented cancer cells against the immune system, blocked them and could improve immunotherapy and help prevent cancer recurrence [62,63].

We examined and screened potential lncRNAs and corresponding microRNAs (miRNAs) that could regulate ALDH2 expression in order to better understand the ALDH2 regulatory network. We discovered that lncRNA PART1 and has-miR-578, as well as has-miR-6833-3p, may be important. By binding and sequestering target miRNAs and participating in mRNA expression regulation, lncRNAs have been shown to act as ceRNAs or molecular sponges in regulating the concentration and biological functions of miRNAs [64,65,66]. A previous study confirmed that the overexpression of lncRNA PART1 promoted cell proliferation, and knockdown of PART1 influenced cell viability and promoted cell apoptosis in PCa [67]. Currently, there is a lack of studies to prove the regulating effect of PART1 and those two miRNAs on ALDH2 expression, we speculated that overexpressed PART1 in PCa could act as a “sponge”, adsorbing and binding has-miR-578 and has-miR-6833-3p, and ultimately regulating the expression of ALDH2 to promote the development and progression of PCa. In addition, we observed that several genes (SIRT3, ENTPD5, GCSH, SLC25A16, TSR1 and FBP1) were predicted to interact with ALDH2 through GeneMania. SIRT3 promoted PCa progression via inhibiting RIPK3-mediated necroptosis and innate immune response [68], while SIRT3-mediated deacetylation of ALDH2 increased enzyme inactivation [69]. FBP1 silencing activated the MAPK pathway, promoting cell epithelial–mesenchymal transition, invasion, and metastasis in PCa [70]. FBP1 and ALDH2 are both involved in a signaling pathway linked to cancer metabolism [71]. However, the tumorigenesis through metabolism of SLC family [72] and the anti-angiogenesis of TSR1 [73] confirmed in other cancers showed potential interactions between each of them and ALDH2. The expression of PCPH/ENTPD5 increased the invasiveness of human PCa cells via a protein kinase C delta-dependent mechanism [74], which was also linked to lipid metabolism. Furthermore, we discovered three drugs, PHA-793887, PI-103, and piperlongumine, that had stronger correlations with ALDH2. Piperlongumine was suggested to increase the activity of recombinant ALDH2 and protect it from inactivation by lipid aldehydes [75]; thus, we speculated that this drug could play a role in the treatment of prostate cancer in the future.

A previous study discovered that 4-HNE’s angiogenesis function via the SIRT3-HIF-1-VEGF axis was associated with hypoxia in TME and resulted in cancer development and immune dysfunction via metabolic reprogramming and metabolic competition [76]. Bipradas Roy et al. [77] later confirmed the finding. As a result, we concluded that ALDH2 may act as an anti-tumorigenesis molecule by not only reducing lipid peroxidation damage, but also promoting angiogenesis in PCa by regulating the amount of 4-HNE. Ferroptosis is a type of programmed cell death characterized by the accumulation of iron-dependent lipid peroxides, in which ROS-induced lipid peroxidation plays an important role [78]. Furthermore, ferroptosis has been linked to the inhibition of the development of multiple cancers [79]. Lower levels of 4-HNE were found to be more sensitive to ferroptosis in lung adenocarcinoma [80], so we hypothesized that decreasing 4-HNE via ALDH2 would cause tumor cells to be more sensitive to ferroptosis in PCa. ALDH2 polymorphisms are also linked to cancer occurrence and progression [18]. The wild and variant alleles are generally referred to as ALDH2*1 and ALDH2*2, resulting in three types of ALDH2 gene: wild-type homozygote (ALDH2*1/*1), heterozygote (ALDH2*1/*2), and variant-type homozygote (ALDH2*2/*2) [81]. Individual enzyme activity in heterozygote individuals (ALDH2*1/*2) was significantly lower (50%) than in the wild type, whereas homozygous mutation enzyme activity (ALDH2*2/*2) was between 1% and 4% of the ALDH2*1/*1 genotype [18]. Thus far, epidemiologic evidence for a link between alcohol consumption and the risk of PCa is lacking. Many studies, meta-analyses, and systematic reviews produced contradictory results [82,83,84,85]. In terms of genes, this condition could be linked to ALDH2 polymorphisms. As a result, detecting the ALDH2 genotype may aid in screening corresponding PCa patients and promoting precision medicine.

Our findings confirmed that ALDH2 repression is linked to a poor prognosis in PCa patients. However, further research into the relationship between ALDH2 and the immune system, as well as the regulation of ALCH2 from various perspectives, such as the ceRNA network and transcription factors, is required. Furthermore, the effect of ALDH2 polymorphisms on cancer progression is not insignificant.

## 5. Conclusions

ALDH2 was discovered to be a potential biomarker for predicting biochemical recurrence in PCa patients. Furthermore, the effect of metabolism reprogramming on TME must be investigated further.

## Figures and Tables

**Figure 1 molecules-27-06000-f001:**
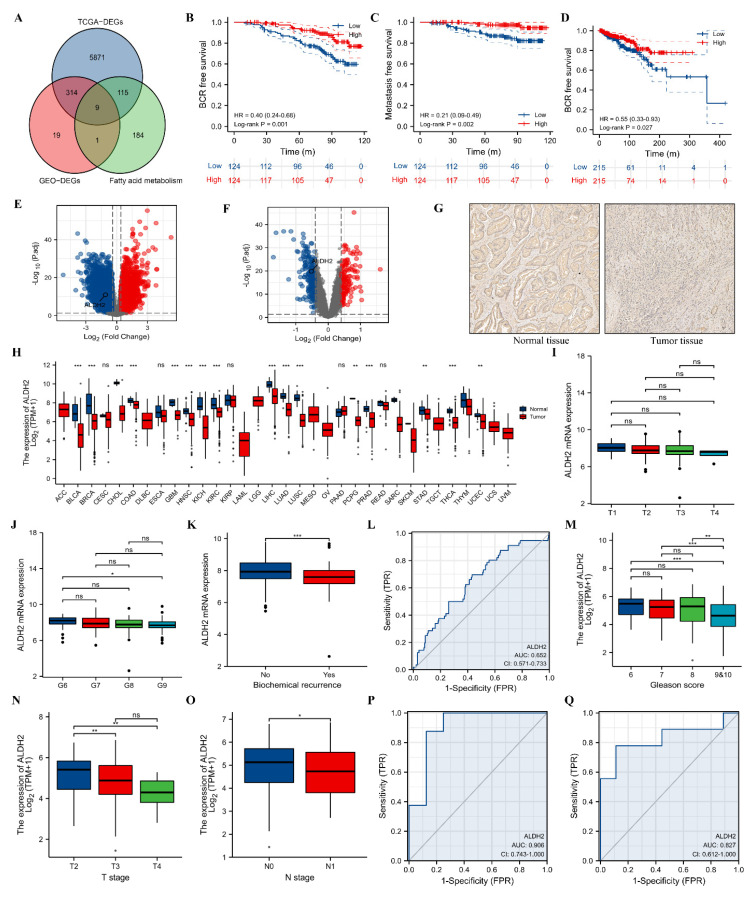
The clinical values of ALDH2 in prostate cancer. (**A**) Venn plot showing through the intersection of DEGs of TCGA database and GEO datasets, and gene sets of fatty acid metabolism; (**B**) Kaplan–Meier curve of ALDH2 for BCR-free survival; (**C**) Kaplan–Meier curve of ALDH2 for metastasis-free survival; (**D**) Kaplan–Meier curve of ALDH2 for BCR-free survival in TCGA database; (**E**) volcano plot showing DEGs in TCGA database; (**F**) volcano plot showing DEGs in GEO datasets; (**G**) differential expression of ALDH2 between tumor and normal tissues at protein level; (**H**) differential mRNA expression of ALDH2 between tumor and normal sample at pan-cancer level in TCGA database; (**I**) comparison between ALDH2 and T stages; (**J**) comparison between ALDH2 and Gleason score; (**K**) comparison between ALDH2 and BCR; (**L**) ROC curve of ALDH2 discriminating BCR from no BCR; (**M**) comparison between ALDH2 and Gleason score in TCGA database; (**N**) comparison between ALDH2 and T stages in TCGA database; (**O**) comparison between ALDH2 and N stages in TCGA database; (**P**) ROC curve of discriminating drug resistance from drug sensitivity; and (**Q**) ROC curve of discriminating radiation resistance from radiation sensitivity. DEG = differentially expressed genes; BCR = biochemical recurrence; ROC= receiver operating characteristic curve. ns, 0.05; *, 0.05; **, 0.01, and ***, 0.001.

**Figure 2 molecules-27-06000-f002:**
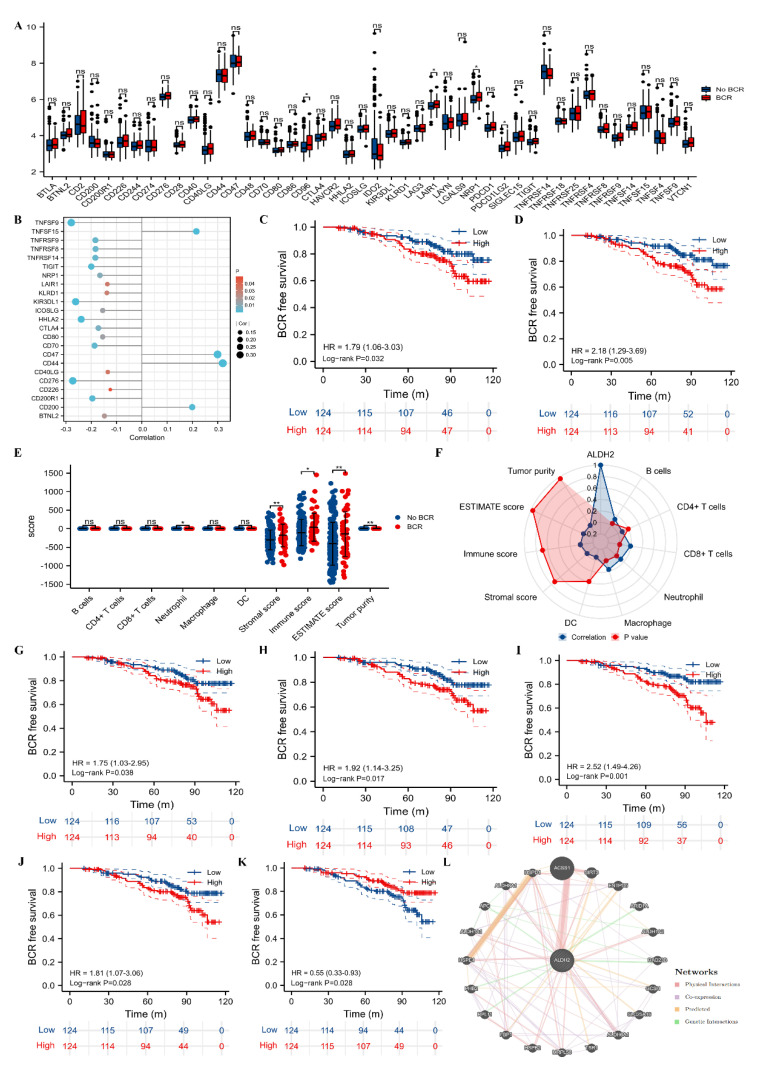
TME analysis and gene interaction with ALDH2. (**A**) Comparison between BCR and no BCR groups for the 46 immune checkpoints; (**B**) bubble plot showing correlations between ALDH2 and immune checkpoints with statistical significance; (**C**) Kaplan–Meier curve of CD96 for BCR-free survival; (**D**) Kaplan–Meier curve of NRP1 for BCR-free survival; (**E**) comparison between BCR and no BCR groups for the TME parameters; (**F**) radar plot showing correlations between ALDH2 and the TME parameters; (**G**) Kaplan–Meier curve of neutrophils for BCR-free survival; (**H**) Kaplan–Meier curve of immune score for BCR-free survival; (**I**) Kaplan–Meier curve of stromal score for BCR-free survival; (**J**) Kaplan–Meier curve of estimate score for BCR-free survival; (**K**) Kaplan–Meier curve of tumor purity for BCR-free survival; and (**L**) network plot showing genes interacted with ALDH2. TME = tumor immune microenvironment; BCR = biochemical recurrence.ns, 0.05; *, 0.05.

**Figure 3 molecules-27-06000-f003:**
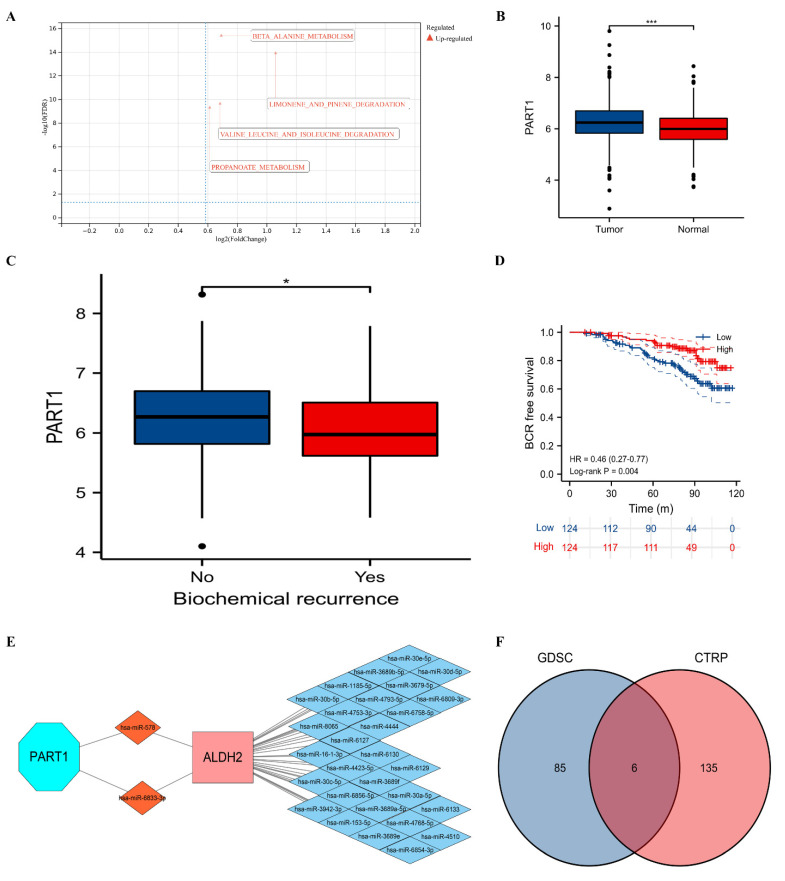
Functional enrichment analysis, ceRNA network, and drug analysis. (**A**) Gene set enrichment analysis; (**B**) comparison between tumor and normal groups for PART1; (**C**) comparison between BCR and no BCR groups for PART1; (**D**) Kaplan–Meier curve of PART1 for BCR-free survival; (**E**) ceRNA network; and (**F**) Venn plot showing the commonly sensitive drugs to ALDH2. ceRNA = competing endogenous RNA; BCR = biochemical recurrence. ns, 0.05; *, 0.05; and ***, 0.001.

## Data Availability

The results shown here are in whole or part based upon data generated by the TCGA Research Network: https://www.cancer.gov/tcga (accessed on 1 November 2021).

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
