# Peer review of "Mitochondrial Aldehyde Dehydrogenase 2 Represents a Potential Biomarker of Biochemical Recurrence in Prostate Cancer Patients"

_molecules, 2022, doi:10.3390/molecules27186000_

Round 1

Reviewer 1 Report

The title can be considered to be revised. "Mitochondrial aldehyde dehydrogenase 2 represents a potential biomarker of biochemical recurrence in prostate cancer patients" can be a candidate.

Fig. 1G shows microscopic images of normal and tumor tissues. The authors should explain this data in the text. Also, the authors need to confirm whether the figure legends are correctly described (line 154). Also, the authors need to include the experimental procedures for this data in Methods, if it is missing.  

The authors should cite Fig. 1L in the text and explain what this ROC curve indicates.

In the abbreviations table, miRNA -> microRNA.

In lines 15 and 131, ADLH2 -> ALDH2

In Fig. 2B, it is unclear if blue and red circles indicate.

In line 181, Fi. 2G-J -> Fig. 2G-K (include K? curve for tumor purity for BCR-free survival)

The authors should cite Fig. 3D in the text and explain what these Kaplan-Meyer curves indicate.

It is recommended that Entire text is subjected to English Editing by native expert.

Author Response

The title can be considered to be revised. "Mitochondrial aldehyde dehydrogenase 2 represents a potential biomarker of biochemical recurrence in prostate cancer patients" can be a candidate.

Response: Thanks for your suggestions. This is a good title and we have changed our title.

Fig. 1G shows microscopic images of normal and tumor tissues. The authors should explain this data in the text. Also, the authors need to confirm whether the figure legends are correctly described (line 154). Also, the authors need to include the experimental procedures for this data in Methods, if it is missing. 

Response: Thanks for your comments. We are sorry for the mistakes. We have explained this data and corrected the related description in the figure legends.

The authors should cite Fig. 1L in the text and explain what this ROC curve indicates.

Response: Thanks for your comments. We are sorry for the mistakes. We have revised cited Fig. 1L and explain in the text.

In the abbreviations table, miRNA -> microRNA.

Response: Thanks for your careful review. We have revised this.

In lines 15 and 131, ADLH2 -> ALDH2

Response: Thanks for your careful review. We have revised this.

In Fig. 2B, it is unclear if blue and red circles indicate.

Response: Thanks for your careful review. We think that the smaller font size leads to this. Thus, we change the presentation of data only presenting the significant immune checkpoints.

In line 181, Fi. 2G-J -> Fig. 2G-K (include K? curve for tumor purity for BCR-free survival)

Response: Thanks for your careful review. We have added the omissive Fig. 2K.

The authors should cite Fig. 3D in the text and explain what these Kaplan-Meyer curves indicate.

Response: Thanks for your careful review. We have added the omissive Fig. 3D.

It is recommended that Entire text is subjected to English Editing by native expert.

Response: Thanks for your comments. We have checked the correspondence and/or figures/tables by a Singapore student who is a native English speaker and now study in our hospital. In fact, all of the authors of this manuscript had checked the wording before the initial submission. Three of the authors had the experience of studying in English speaking country and already had many lectures published. We also used AJE service (https://www.aje.com/go/sncnh) to improve our manuscript through our institutional portal. We tried our best to correct the wording.

Reviewer 2 Report

1. The authors have studied a large amount of data meticulously and have come to some very intersting conclusions. The authors can highlight the uniqueness and mechanistic advance of their study in the background section of the manuscript. The findings are important and have the potential for future explorations in stemness and multidrug resistance.

2. Please mention the molecules stained in Fig. 1G. Mention of Fig. 1G is missing from the result section of the manuscript.

3. Fig 2B and 2L label font sizes are too small to be read. It would be helpful if the labels and font sizes in the figures were of uniform size.

4. The authors could describe the role of amino metabolism in tumor infiltration in the background section of the manuscript and its importance in the results or discussion section

5. "So far, epidemiologic evidence for association between alcohol intake and the risk of PCa still remain unclear" it does not seem this statement is important for the present study. 

Author Response

  1. The authors have studied a large amount of data meticulously and have come to some very interesting conclusions. The authors can highlight the uniqueness and mechanistic advance of their study in the background section of the manuscript. The findings are important and have the potential for future explorations in stemness and multidrug resistance.

Response: Thanks for your comments. We have added your constructive suggestions in the background section.

  1. Please mention the molecules stained in Fig. 1G. Mention of Fig. 1G is missing from the result section of the manuscript.

Response: Thanks for your comments. We are sorry for the mistakes. We have explained this data and corrected the related description in the figure legends.

  1. Fig 2B and 2L label font sizes are too small to be read. It would be helpful if the labels and font sizes in the figures were of uniform size.

Response: Thanks for your careful review. We change the presentation of data only presenting the significant immune checkpoints and reorder the figure 2.

  1. The authors could describe the role of amino metabolism in tumor infiltration in the background section of the manuscript and its importance in the results or discussion section

Response: Thanks for your suggestion. In this study, we mainly focused on fatty acid metabolism. Thus, we think you might mean the fatty acid metabolism. We have added related contents in background and discussion.

  1. "So far, epidemiologic evidence for association between alcohol intake and the risk of PCa still remain unclear" it does not seem this statement is important for the present study.

Response: Thanks for your comments. Because ALDH2 was associated with alcohol metabolism and several study showed that alcohol intake was associated with PCa risk. Thus, we added this description to indicate that ALDH2 warranted more detailed studies.
